

# Longitudinal effects of aerobic training programme on body composition in non-elite adolescent female swimmers

Mariusz Kuberski[1], Agnieszka Musial[2], Maciej Choroszucho[1],
Jan M. Konarski[3] and Jacek Wąsik[1]

[1] Institute of Physical Culture Sciences, Jan Długosz University in Częstochowa, Częstochowa, Śląskie, Poland
[2] Institute of Psychiatry, Psychology & Neuroscience, Social, Genetic and Developmental Psychiatry Centre, King's College London, London, United Kingdom
[3] Department of Theory of Sports, Poznań University of Physical Education, Poznań, Wielkopolskie, Poland

Corresponding author
Mariusz Kuberski,
m.kuberski@ujd.edu.pl

## ABSTRACT

**Introduction:** The aim of this study was to assess the impact of a 3-year swimming training programme on body fat measurements in adolescent girls, without prior selection.

**Method:** Two groups of 10-year-old girls were analysed at the beginning of the study (4th grade in primary school). The experimental group consisted of 14 swimmers (body mass: 34.99 ± 2.77 kg; height: 146.00 ± 3.05 cm). The control group consisted of 14 girls (body mass: 37.93 ± 6.02 kg; height: 145.55 ± 3.88 cm) who only participated in mandatory physical education classes. The study was conducted over a period of 3 years, with measurements taken every 6 months. Body fat was measured through the thickness of skinfolds at four anatomical locations: above the biceps; above the triceps; below the lower angle of the scapula; and above the superior iliac crest. Based on these measurements, the percentage of body fat was calculated.

**Results:** Statistical analysis revealed that despite the lack of initial selection, there was no significant difference in the percentage of body fat between the experimental and control groups at the start of the study—18.62% *vs* 24.85%. This difference persisted until the final measurement after 3 years, at which point it became statistically significant—17.31% *vs* 27.14% (F = 37.44, $p < 0.05$).

**Discussion:** Our findings indicate that 3 years of swimming training in adolescent girls resulted in a reduction in fat tissue growth across all four measurement sites. Initially, body fat percentage had a strong negative impact on $VO_2$ max, particularly in the experimental group. However, this effect diminished in both groups over the course of the study.

**Conclusion:** Swimming may be an effective means of preventing obesity. Coaches should consider a slim physique in swimming candidates.

## INTRODUCTION

### The impact of swimming on the human body

By conducting a meta-analysis of scientific studies, it can be clearly stated that regular physical activity has a positive effect on the human body (*Welis, Yendrizal & Tri Mario, 2023*; *Pippi et al., 2024*). In Poland, it has been shown that the physical activity levels of girls and boys aged 11–17 fall below the standards recommended by the World Health Organisation (WHO) (*Guthold et al., 2020*). Consequently, it is advised that children and adolescents between the ages of 5 and 17 engage in moderate-intensity aerobic exercise for approximately 60 minutes per day, as such activity positively influences biological development (*Malina et al., 2011*; *Kuberski, Góra & Wąsik, 2024*; *Bunc, 2022*). Swimming, a low-impact sport with numerous benefits, is an ideal choice to meet these guidelines. Numerous studies have demonstrated that swimming yields substantial physiological benefits (*e.g.*, improved physical endurance), psychological benefits (*e.g.*, increased self-confidence), and social benefits (*e.g.*, interaction with peers), attributed to structured training (*Kuberski, Góra & Wąsik, 2024*; *Silva et al., 2020*; *Shoemaker et al., 2019*; *Kuberski, Musial & Choroszucho, 2025*). Compared to other sports, swimming poses a low injury risk due to the reduced strain on joints as a result of swimming in a horizontal position (*Kuberski, Musial & Choroszucho, 2025*; *Morais et al., 2021*). By engaging large muscle groups and minimizing the load on joints and the spine, swimming is frequently recommended for rehabilitation purposes and is thus one of the most frequently advocated physical activities across all age groups (*Ondrušová & Koláriková, 2016*). This is due to its involvement of multiple muscle groups simultaneously, contributing to the development of overall muscle strength (*Vaneckova & Kabesova, 2022*).

### Swimming and aerobic capacity

Researchers suggest that swimming training should begin before the onset of puberty to prepare an appropriate base of aerobic capacity for further development (*Lätt et al., 2009*). Swimming competition is divided into four swimming styles (freestyle, backstroke, butterfly, breaststroke) and race distances (from 50 to 1,500 m), highlighting the importance of overall endurance preparation at a high level (*Eider, 2015*). Research findings on oxygen efficiency in children indicate that girls aged 12–15 who engage in swimming activities exhibit higher aerobic capacity than their non-swimming peers (*Roels et al., 2005*). The average increase in $\dot{V}O_2$ max in children aged 11 to 15 who swim is approximately 1.53 litres (*Doherty & Dimitriou, 1997*). However, some scientific reports suggest a decrease in endurance levels in both girls and boys training swimming compared to control groups (*Wieczorek, 2001*). The authors attribute this decline to general hormonal changes in the body, resulting in increased body mass and fat tissue accumulation (*Jürimäe et al., 2007*). It is important to note that the aforementioned studies, both those showing increased aerobic capacity and those showing a decline, focused on individuals selected for competitive swimming.

## Consequences of obesity

Body fat percentage is a heritable trait, influenced by external factors (*Brook, Huntley & Slack, 1975*; *Cardon et al., 1994*), primarily physical activity and diet. Aerobic exercise plays a crucial role in maintaining health and biological fitness. Numerous authors stress that insufficient physical activity can lead to cardiovascular diseases, diabetes, and obesity, as well as certain types of cancer (*Boreham et al., 2001*; *Weinstein et al., 2004*). A low level of physical activity combined with the excessive consumption of processed foods contributes to the accumulation of body fat (*Varela-Garrote et al., 2022*). Over the last several years, there has been a growing concern regarding the declining physical activity levels in children and adolescents, particularly due to prolonged screen time and numerous virtual extracurricular activities. Behavioural patterns learned in childhood tend to persist into adulthood, with an increasing number of people working sedentary jobs and not participating in physical activities (*Kohl & Hobbs, 1998*). This issue has been highlighted by the World Health Organisation, which issued recommendations regarding physical activity levels for various age groups (*Guthold et al., 2020*). The health risks associated with excessive body fat are particularly dangerous for young women who may soon plan to become mothers. Obesity may increase the risk of infertility (*Kirchengast & Huber, 2001*), as well as metabolic diseases in the fetus and pregnancy complications (*Cedergren, 2004*). The authors of other studies suggest that for additional activity to produce the desired results and significantly impact health benefits, it must be supported by an appropriately tailored diet, especially considering the age of the study participants (*Charmas & Gromisz, 2019*). In other studies, some authors have shown that halting the increase in body fat supports the overall balance of the body by producing many biologically active substances known as adipokines, which ensure the proper functioning of the immune, endocrine, and metabolic systems (*Agrawal, Kern & Nikolajczyk, 2017*; *Poddar, Chetty & Chetty, 2017*; *Boengler et al., 2017*). They also emphasise that swimming significantly reduces fat deposition in both the upper and lower body.

## Preliminary selection in swimming

In the selection process of children for competitive swimming, determining biological age—indicating the level of organism development—is crucial. Differences in biological age among peers can range from 3 to 4 years, or even more during puberty (*Eider, 2015*). Often, early maturing individuals achieve high junior-level athletic results that do not translate into senior-level performance. Hence, some authors recommend selecting individuals with average or slow maturation rates for competitive swimming, ensuring they progress through the necessary stages of training (*Lätt et al., 2009*).

An important criterion in the initial selection process is assessing specific predispositions for competitive swimming, such as good balance and buoyancy, the ability to adopt a streamlined body position underwater, and a wide range of motion, especially in the shoulder joints (*Jagomägi & Jürimäe, 2005*). Another criterion is body composition,

which significantly impacts swimming performance, particularly body height and proportions between different body segments, as noted by various authors (*Cortesi et al., 2020*; *Alves et al., 2022*). It is assumed that the height of candidates for competitive swimming should be at least one standard deviation above the average for children of the same age group, and this advantage should persist into adulthood (*Lätt et al., 2010*). Significant characteristics that influence swimming performance include shoulder width, an upper body build with a narrow pelvis, and slender hips (*Kuberski, Góra & Wąsik, 2024*). Research conducted on 18–19 year old swimmers also highlights the importance of having a V-shaped torso (broad shoulders, narrow hips), longer upper limbs and large hands as predictors of swimming performance (*Rejman et al., 2023*). This body composition facilitates faster movement in the aquatic environment. Some authors emphasise that prepubescent girls should have a slim figure, a long torso, narrow shoulders, a narrow pelvis, a deep chest, and a low body mass relative to height for more effective swimming (*Cochrane et al., 2015*). They also indicate that excess fat tissue increases resistance in water, particularly when swimming underwater after a starting dive or turn (*Cochrane et al., 2015*).

## Purpose and hypotheses

Based on the literature, it can be concluded authors are not unanimous about the effects of competitive swimming on physiological characteristics in young girls. The present study aimed to evaluate the aerobic capacity and body fat of adolescent girls as influenced by swimming training. It was hypothesised that swimming reduces the body fat of the adolescent girls studied. Taking into account the fact of natural biological development occurring in children of this age, it is reasonable to track the changes resulting from swimming in long-term studies to which people will be recruited without preliminary selection.

The aim of this study was to assess the effect of a 3-year swimming training program on body fat in adolescent girls who were not selected for swimming based on their body composition.

Therefore, the following research questions were asked: What are the differences in the level of body fat between girls who regularly trained swimming and those who did not swim? How did the group of swimmers differ in terms of aerobic capacity from the control group after 3 years of swimming training? To what extent does body fat affect aerobic capacity in both groups?

The results may help physical education teachers, coaches, sports instructors and physiotherapists to determine the extent to which swimming can help combat early obesity, given the low risk of injury associated with the aquatic training environment.

## MATERIALS AND METHODS

### Participants

The study involved two groups of girls, with each group consisting of 14 participants. The experimental group included swimmers (mean biological age: 10.52 ± 0.37 years, mean

body weight: 34.99 ± 2.77 kg; mean height: 146.00 ± 3.05 cm at the time of the research commencement) who trained at student sports clubs in Czestochowa, Poland. Recruitment for the sports clubs occurred without any initial pre-selection which means that the swimming group consists of girls who voluntarily decided to participate in swimming training. Therefore, the choice of people was not dictated by the children's predispositions, physical parameters or talent. Upon starting the study, the girls began swimming training, though they had prior swimming skills from attending swimming lessons twice a week. The control group consisted of girls (mean biological age: 10.74 ± 0.62 years, mean body weight: 37.93 ± 6.02 kg; mean height: 145.55 ± 3.88 cm at the time of the research commencement) who only participated in mandatory physical education classes. It is important to note than according to information obtained from their legal guardians, apart from their obligatory physical education lessons, girls from the control group did not participate in any sports training and all girls from both groups were premenarchal, during all six stages of the study.

## Ethics

In accordance with the requirements of the Declaration of Helsinki, all participants and their parents were informed about the purpose and methodology of the research. They provided written consent for participation, and the protocol of the study was approved by the Bioethics Committee for Scientific Research at the Jan Dlugosz University in Czestochowa, approval number KB-2/2012.

## Protocol

The research project was of the experimental type with a longitudinal character.

The study was conducted over a period of three consecutive years—between autumn 2011 and spring 2014, with measurements taken every 6 months between 8 am and 12 pm (six total measurements). In both groups, body weight and height were measured using a scale with a stadiometer (WPT 150,0; RadWag; Polska), with a precision of 0.1 kg and 0.5 cm, respectively. The developmental stage, or biological age, of the girls was calculated using the following formula (*Przewęda, 1971*).

$$\text{Biological age} = \frac{\text{body mass age} + \text{body height age} + \text{chronological age}}{3}$$

where body mass age and body height age were estimated using Pirquet's tables for girls from the Lubusz region (*Malinowski et al., 2005*). Chronological age was calculated as the time between the participant's date of birth and the date of measurement (*Jopkiewicz & Suliga, 1998*).

Body fat percentage was assessed by measuring skinfold thickness at four anatomical sites: over the biceps, over the triceps, beneath the inferior angle of the scapula (referred to as shoulder blade), and above the iliac crest (referred to as stomach). All measurements were taken on the right side of the body in a standing position (Frankfurt plane) using a Harpenden skinfold caliper (M2 TOP; Käfer, Germany) with a precision of 0.1 mm. Based on the skinfold measurements, the body fat percentage was calculated using the formula

provided by *Slaughter et al. (1988)*. If the sum of the triceps and subscapular skinfolds was ≤35 mm, the equation used was:

% Body fat $= [1.33 \times (R + L)] - [0.013 \times (R + L)^2] - 2.5.$

If the sum of the triceps and subscapular skinfolds was >35 mm, the equation used was:

% Body fat $= 0.546 \times (R + L) + 9.7$

where R represents the skinfold measurement over the triceps, L represents the skinfold measurement beneath the scapula.

The Maximal Multistage 20-m Shuttle Run Test (commonly referred to as the Beep Test) was used to measure aerobic capacity (*Kasai et al., 2023*). This test involved running back and forth over a 20-m distance, with the pace controlled by audio signals. The participants had to complete the distance within the time allotted by the sound, which gradually shortened with each subsequent stage. The initial running speed was 8.5 km/h, and it increased by 0.5 km/h at each stage. The number of shuttle runs also increased with each stage: the first stage required seven runs at a consistent pace, the second stage required eight runs, and so on, up to the fourth stage. From stages five to eight, participants completed 10 runs, and from stages nine to thirteen, 12 runs of 20 m were required. If a participant failed to reach the line before the next sound, their attempt was terminated. The total number of successful shuttle runs was recorded. Based on the speed at which the last stage was completed and the participant's calendar age, maximal oxygen uptake ($VO_2$ max), an indicator of aerobic fitness, was calculated using the formula provided by *Léger et al. (1988)*.

$\dot{V}O_2$ max $= 31.025 + 3.238 \times P - 3.248 \times W + 0.1536 \times P \times W$

where P represents the maximum running speed (km/h) from the last completed stage. W represents the calendar age, rounded down to the nearest whole number.

## Swimming training

The swimmers' training macrocycle was designed based on the British Swimming Federation guidelines for girls aged 9–12 years (*Lang & Light, 2010*), consisting of four training sessions per week, held in the morning (6:30–7:40 am). Each training session lasted 70 min, with an 80% to 20% ratio of aerobic to anaerobic exercises. The average daily swimming distance was approximately 1,500 m in the first year, 2,000 m in the second year, and 2,500 m in the third year. Each session began with a 10-min warm-up on land and a warm-up in the water—swimming from 200 to 400 m in the front crawl or back crawl. The main part of the training session, lasting 30 min, was devoted to perfecting the technique of all swimming styles and developing endurance capabilities, where swimmers performed several series of sections (Table 1). Attention was paid to body position, effectiveness of arm strokes and leg work, maintaining the correct "swimming step". Turns and swimming under the water surface were improved. To improve technical elements, coaches used specialist swimming equipment, including: fins (short and long), swimming paddles and specialist rubbers. Aerobic capacity was usually shaped by swimming the
**Table 1 Swimming training plan for 3 years of research.**

| Research time | Number of trainings | Training unit diagram | Average distance |
|---|---|---|---|
| 1 year (35 weeks) | 4 training sessions per week | Warm up: 200–300 m<br>Main part:<br>5 × 50 m only arms<br>5 × 50 m only legs<br>5 × 50 m coordination arms/legs<br>2 × 100 m full style<br>Warm down: 200–300 m | 1,500 m in 1 training session |
| 2 year (35 weeks) | 4 training sessions per week | Warm up: 300–400 m<br>6 × 50 m only arms<br>6 × 50 m only legs<br>6 × 50 m coordination arms/legs<br>5 × 100 m full style<br>Warm down: 200–300 m | 2,000 m in 1 training session |
| 3 year (35 weeks) | 4 training sessions per week | Warm up: 300–400 m<br>5 × 100 m only arms<br>5 × 100 m only legs<br>5 × 100 m coordination arms/legs<br>4 × 100 m full style<br>Warm down: 200–300 m | 2,500 m in 1 training session |

crawl on the chest. The training unit ended with stretching exercises on land and lasted about 7 min. Appropriate stretching exercises increased the range of motion in the shoulder girdle, as well as the mobility of the ankle joints.

According to a participant classification framework proposed by *McKay et al. (2022)*, the swimming group of this study can be classified into Tier 2: Trained/Developmental.

## Limitations

The study groups were relatively small (14 subjects). After data collection, the sample size estimated using G*Power software (version 3.1.9.2; University of Cologne, Cologne, Germany) returned a minimum of 12 measurement items for: a = 0.05, effect size f = 0.6, and β = 0.95. The diet of the subjects was not taken into account. Aerobic capacity was measured indirectly using a running test due to the lack of consent from the children's legal guardians for direct tests. The assessment of menstrual age obtained from the interview of legal guardians.

## Analyses

The measure of the asymmetry of a distribution (skewness) was used to assess the normality of the data distribution.

To assess the significance of differences in biological age, $VO_2$ max and measures of body adiposity we performed a two-way analysis of variance (ANOVA), comparing measurements at first and sixth timepoint within the experimental and control groups, as

well as between the groups. In order to determine whether measurement timepoint and swimming training interact in influencing changes in aerobic capacity and adiposity features we also examine their interaction. All analyses were conducted using stats for R (*R Core Team, 2021*).

To analyze the longitudinal influence of body fat percentage on changes in aerobic capacity, we employed linear mixed-effects models (LMM) to account for repeated measurements within the sample. Separate models were fitted for each group (experimental and control), allowing for comparison of the effects within and between groups.

The models were structured as follows:

$$\text{VO}_2 \text{ max} = \beta0 + \beta1 \,(\text{Body Fat}_{it}) + \beta2 \,(\text{Time}_t) + \beta3 \,(\text{Body Fat}_{it} \times \text{Time}_t) + u_i + \in_{it}$$

where $\beta0$ represents the intercept (baseline aerobic capacity); $\beta1$, $\beta2$ and $\beta3$ represent the fixed effects for body fat percentage, time, and their interaction, respectively; $u_i$ represents the random intercept for each participant to account for individual variability; $\in_{it}$ is the residual error.

To ensure the robustness of our LMM results, we conducted a thorough verification of key model assumptions. Normality of residuals was assessed through visual inspection of Q-Q plots, histograms, and skew statistics, with the Shapiro-Wilk test yielding a *p*-value greater than 0.05, indicating no significant deviation from normality. The random effects structure was tested by comparing models with random intercepts and random slopes, and likelihood ratio tests and AIC comparisons showed no significant improvement with random slopes (AIC = 233.95 for both models). Therefore, for parsimony, we retained the model with random intercepts only. We also assessed residual variance for homoscedasticity and confirmed no issues, and variance inflation factors (VIF) indicated no multicollinearity, ensuring the independence of predictors.

The models were fitted using the lme4 (*Bates et al., 2015*) package in R (*R Core Team, 2021*). Marginal R2 (proportion of variance explained by fixed effects) and conditional R2 (proportion of variance explained by both fixed and random effects) were computed using the MuMIn (*Bartoń, 2024*) package to assess model fit. The beta coefficients (fixed effects estimates) were extracted to quantify the effects of body fat percentage and time on $\text{VO}_2$ max. Standard errors were reported for each estimate. Within the experimental group model, the $\text{VO}_2$ max of the swimmers was modelled as a function of body fat percentage, time, and the interaction between body fat and time, with random intercepts for each individual swimmer. The $\text{VO}_2$ max of the non-swimmers was modelled using the same structure to allow for comparison with the experimental group.

## RESULTS

### Descriptive statistics

Upon visual examination of histogram plots and the skew statistics, all variables were normally distributed (Fig. 1 and Table 2). Trajectories of biological age, body fat, aerobic capacity and skinfold thickness measurements throughout the duration of the study are

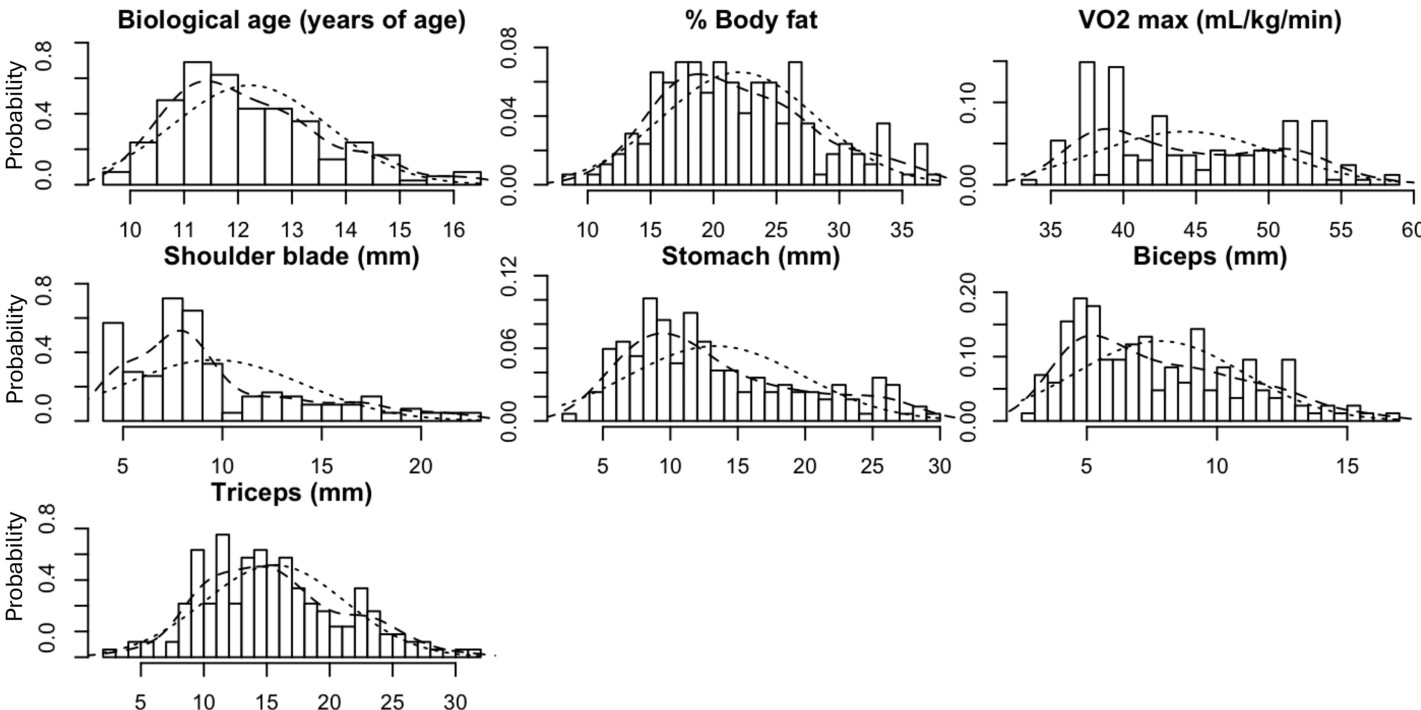

**Figure 1  Histogram plots showing the distribution of the outcome variables.**

illustrated in Fig. 2 and Table 2 and compared between the experimental and control groups.

The results indicated that both at the onset of the study and after 3 years of training, the swimmers did not differ in biological age from their non-swimming peers (Fig. 2 and Table 2). The biological age of the girls increased steadily across both groups throughout the measurement period (Fig. 2). The group of girls engaged in competitive swimming for 3 years significantly differed in skinfold thickness from the non-training group. The analysis of variance revealed that the experimental group exhibited significantly less body fat at all four measurement sites during the initial measurement (Table 2). After 3 years of training, significant differences between the two groups persisted and increased with the mean differences between the initial and final measurements of 2.73 mm at the shoulder blade, 5.56 mm at the stomach area, 3.58 mm above the biceps, and 2.52 mm above the triceps (Table 2). Despite the absence of preliminary selection, the swimmers differed substantially, although not significantly, from the control group in terms of body fat percentage at the initial stage of the study (18.62% *vs* 24.85%), and this difference persisted through to the final measurement after 3 years, when it became statistically significant (17.31% *vs* 27.14%) (Table 2). The VO$_2$ max index significantly differed between the groups, with swimmers demonstrating greater aerobic capacity throughout the study period compared to non-swimmers (Fig. 2 and Table 2). Although there was already a significant difference in aerobic capacity between the groups at the beginning of the measurements (47.48 *vs* 40.15 mL/kg/min), over the 3-year training period, we observed

**Table 2 Descriptive statistics of the first and sixth measurement timepoint, along with results of the ANOVA.**

| Variable | Group | Time 1 | | | | | Time 6 | | | | | ANOVA | | | | | |
|---|---|---|---|---|---|---|---|---|---|---|---|---|---|---|---|---|---|
| | | M | SD | Min | Max | S | M | SD | Min | Max | S | Term | F | P | Eta$^2$ | d E | d C |
| Biological age | Experimental | 10.52 | 0.37 | 9.98 | 11.06 | −0.11 | 13.86 | 1.05 | 12.51 | 16.37 | 0.75 | Fup | 249.88 | 0.00 | 0.82 | −4.23 | −4.23 |
| | Control | 10.74 | 0.62 | 9.86 | 11.88 | 0.26 | 14.43 | 1.07 | 13.05 | 16.44 | 0.49 | G | 3.13 | 0.08 | 0.01 | | |
| | | | | | | | | | | | | Fup × G | 0.63 | 0.43 | 0.00 | | |
| % Body fat | Experimental | 18.62 | 4.50 | 10.77 | 26.75 | 0.17 | 17.31 | 3.45 | 11.38 | 22.56 | 0.05 | Fup | 0.14 | 0.71 | 0.00 | 0.33 | −0.41 |
| | Control | 24.85 | 6.22 | 15.73 | 36.29 | 0.18 | 27.14 | 5.04 | 20.54 | 36.07 | 0.46 | G | 37.44 | 0.00 | 0.41 | | |
| | | | | | | | | | | | | Fup × G | 1.89 | 0.18 | 0.02 | | |
| VO$_2$ max | Experimental | 47.48 | 3.47 | 41.51 | 53.45 | 0.28 | 51.30 | 3.39 | 43.77 | 55.56 | −0.76 | Fup | 0.61 | 0.44 | 0.00 | −1.11 | 1.19 |
| | Control | 40.15 | 1.80 | 39.12 | 43.90 | 1.21 | 37.53 | 2.52 | 33.30 | 42.86 | 0.38 | G | 188.26 | 0.00 | 0.73 | | |
| | | | | | | | | | | | | Fup × G | 17.52 | 0.00 | 0.07 | | |
| **Skinfold thickness indices** | | | | | | | | | | | | | | | | | |
| Shoulder blade | Experimental | 7.85 | 2.41 | 4.60 | 12.50 | 0.18 | 6.86 | 1.82 | 4.00 | 9.10 | −0.38 | Fup | 0.13 | 0.72 | 0.00 | 0.46 | −0.34 |
| | Control | 11.20 | 5.18 | 5.30 | 21.20 | 0.65 | 12.94 | 4.94 | 7.40 | 22.50 | 0.40 | G | 20.57 | 0.00 | 0.28 | | |
| | | | | | | | | | | | | Fup × G | 1.72 | 0.20 | 0.02 | | |
| Stomach | Experimental | 9.43 | 3.70 | 5.30 | 19.90 | 1.42 | 8.96 | 3.98 | 2.10 | 14.80 | −0.06 | Fup | 3.02 | 0.09 | 0.03 | 0.12 | −0.86 |
| | Control | 14.12 | 5.32 | 6.90 | 22.50 | 0.21 | 19.21 | 6.41 | 11.00 | 29.90 | 0.25 | G | 31.57 | 0.00 | 0.35 | | |
| | | | | | | | | | | | | Fup × G | 4.36 | 0.04 | 0.05 | | |
| Biceps | Experimental | 7.44 | 3.13 | 4.10 | 13.00 | 0.68 | 5.37 | 1.85 | 3.10 | 8.20 | 0.12 | Fup | 0.13 | 0.72 | 0.00 | 0.80 | −0.49 |
| | Control | 9.00 | 3.28 | 5.20 | 14.50 | 0.35 | 10.51 | 2.96 | 3.60 | 16.70 | −0.20 | G | 19.26 | 0.00 | 0.25 | | |
| | | | | | | | | | | | | Fup × G | 5.48 | 0.02 | 0.07 | | |
| Triceps | Experimental | 12.31 | 4.11 | 5.50 | 19.50 | 0.09 | 11.47 | 3.43 | 4.20 | 16.10 | −0.41 | Fup | 0.12 | 0.73 | 0.00 | 0.22 | −0.32 |
| | Control | 18.16 | 5.99 | 7.60 | 27.50 | −0.13 | 19.84 | 4.53 | 13.60 | 31.10 | 0.83 | G | 33.24 | 0.00 | 0.38 | | |
| | | | | | | | | | | | | Fup × G | 1.05 | 0.31 | 0.01 | | |

**Note:**
G, Group; M, mean; SD, standard deviation; S, skew; Fup, follow-up; d, Cohen's d.; E, experimental group; C, control group.

the interaction between the effects of swimming training and measurement timepoint, with the swimming group experienced a significant increase in VO$_2$ max, while the non-training group showed a significant decline in this parameter (51.30 *vs* 37.53 mL/kg/min) (Fig. 2 and Table 2).

Regression analyses predicting VO$_2$ max from body fat percentage across six follow-up points demonstrated distinct patterns for the experimental (swimming) and control (non-swimming) groups (Fig. 3). The LMM analysis revealed that, across the six follow-ups, percentage of body fat explained more variation in aerobic capacity in the experimental group, compared to the control group. The marginal R2 values were estimated as 15% in the group of swimmers and 12% in controls, whereas the conditional R2 values were estimated as 66% and 82%, respectively. This discrepancy suggests that the inclusion of random effects substantially altered the predictive power of the fixed effects in the model.

The linear regression analyses performed for each individual measurement timepoint revealed distinct patterns for swimmers and non-swimmers. At the initial measurement, swimmers exhibited a strong negative relationship between body fat percentage and

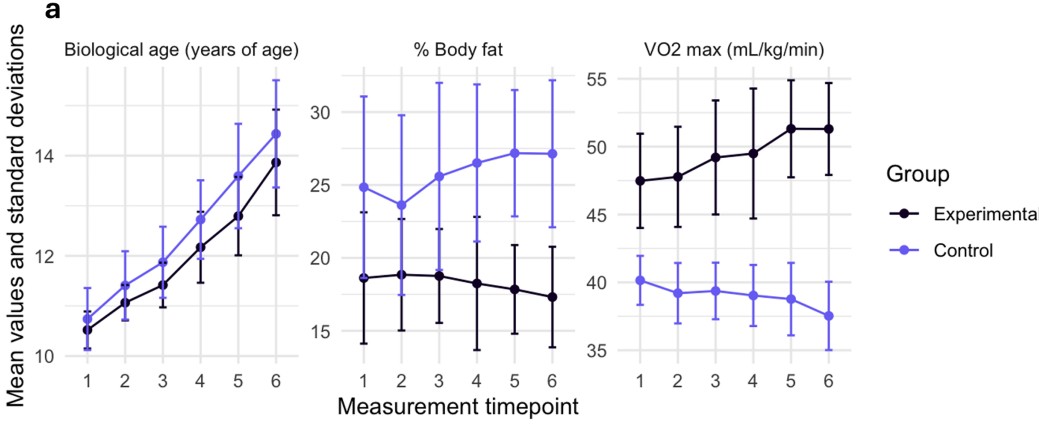

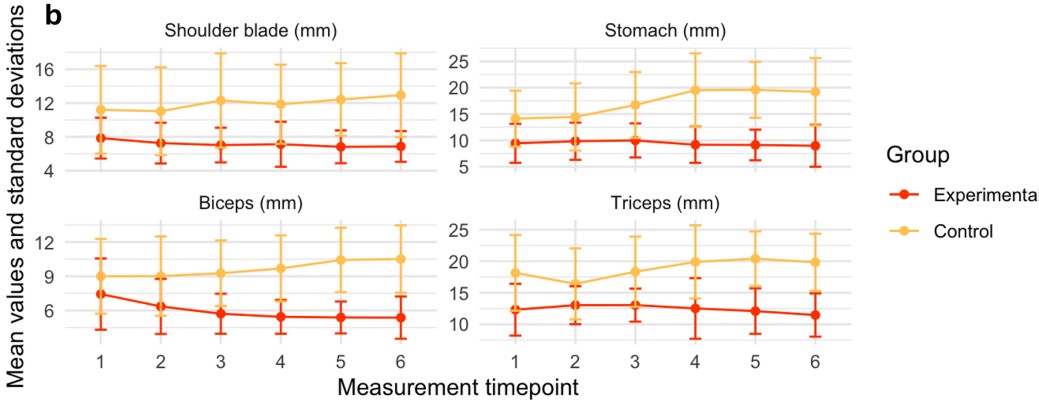

**Figure 2** Mean trajectories of biological age, body fat and aerobic capacity ($VO_2$ max) measures (A) and body adiposity measurements (B) across 3-years of follow-ups.

$VO_2$ max, explaining 21.8% of the variance (Beta = −0.35, SE = 0.19). In the control group, the effect was much weaker, predicting 2% of the variance (Beta = −0.04, SE = 0.08). By the second follow-up, the negative impact of body fat percentage on $VO_2$ max in swimmers attenuated, with a beta of −0.24 (SE = 0.28) and R2 reduced to 6%. In the control group, the relationship strengthened, with a beta of −0.17 (SE = 0.09) and R2 of 24%. In subsequent follow-ups, the predictive power of body fat continued to weaken for both groups. By the sixth follow-up, the relationship between percentage of body fat and aerobic capacity was nonsignificant in both groups. Overall, while the percentage of body fat initially had a strong negative effect on $VO_2$ max, especially for swimmers, its influence diminished over time.

## Effect size measures: eta$^2$ and Cohen's d

The eta$^2$ statistic provides an estimate of the proportion of variance in the dependent variable explained by the independent variable (*Richardson, 2011*). Higher eta$^2$ values indicate a greater proportion of variance explained. Based on commonly accepted benchmarks (small: 0.01, medium: 0.06, large: 0.14), the eta$^2$ values in this study suggest

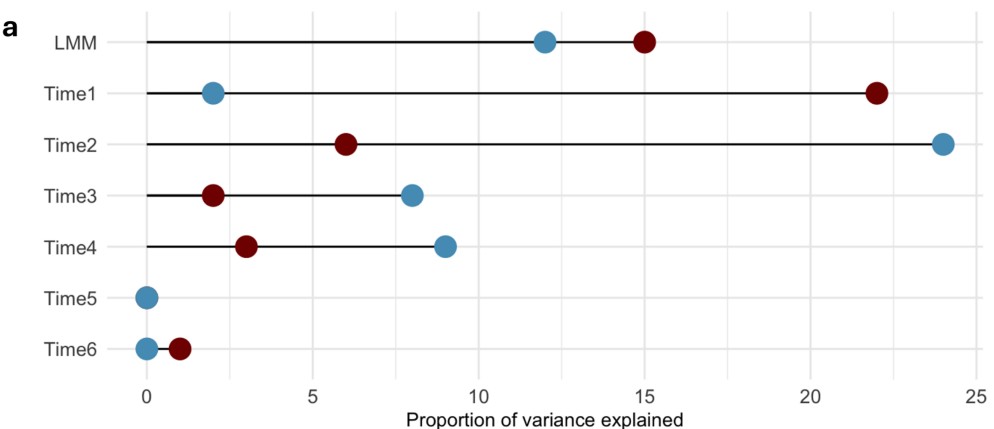

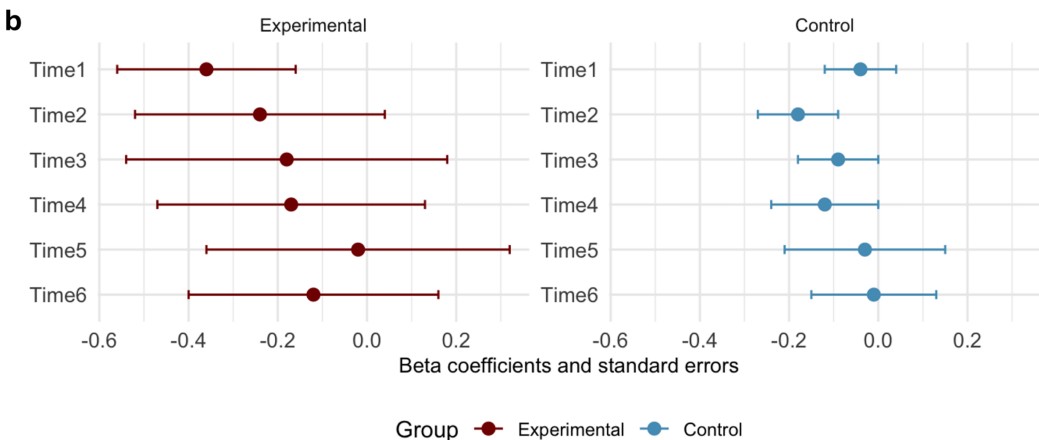

**Figure 3 Results of the regression analyses.** (A) The proportion of variance explained ($R^2$) by the percentage of body fat across the 3-year study period, using the linear mixed model approach (LMM). The remaining rows—$R^2$ values estimated using linear regressions, predicting $VO_2$ max from body fat percentage. (B) Beta coefficients for the experimental and control groups.

strong effects for biological age ($\eta^2 = 0.82$) and $VO_2$ max ($\eta^2 = 0.73$), indicating that training had a substantial impact on these variables. Similarly, body fat percentage showed a large effect ($\eta^2 = 0.41$), while skinfold thickness indices exhibited moderate to large effects, ranging from $\eta^2 = 0.25$ to $0.38$, depending on the site (Table 2).

Cohen's d quantifies the standardized mean difference between groups (*Cohen, 2013*). According to conventional guidelines (small: 0.2, medium: 0.5, large: 0.8), the effect size for $VO_2$ max at the final measurement was $d = 1.19$, indicating a large effect of swimming training on aerobic capacity. Similarly, effect sizes for skinfold thickness ranged from $d = -0.32$ to $-0.86$, reflecting small to large reductions in body fat at different measurement sites. For body fat percentage, Cohen's d was $-0.41$ at the final time point, suggesting a small to moderate effect. These interpretations align with established benchmarks, reinforcing the meaningful differences observed between groups (Table 2).

## DISCUSSION

Our study supports our hypothesis that 3 years of swimming training reduces the body fat of the adolescent girls studied. The present research indicates that the biological age of both studied groups remained within the normative range throughout the experiment. No significant changes in this index were observed as a result of the 3-year swimming training in the experimental group, suggesting that this type of physical activity did not accelerate or delay the biological development of girls aged 10–12 years. The findings regarding biological development in children who engage in swimming compared to their peers who do not participate in additional physical activity are inconclusive (*Benefice & Malina, 1996*). The studies of *Wawrzyniak (2001)* demonstrated that children who trained in swimming exhibited a higher degree of biological maturity compared to their non-swimming peers. However, it should be noted that this outcome could be attributed to prior selection for swimming. Similarly, *Wieczorek & Witkowski (1990)* suggested that the higher level of biological development among third-grade boys who train swimming may reflect effective selection processes for this sport. Conversely, research by *Nowacka-Chiari (2005)*, which included girls aged 11–12, did not reveal significant differences in biological development between those who trained in swimming and those who did not engage in additional physical activity. It is important to recognise that an accurate estimation of biological age allows for the individualisation of training loads to match the developmental stage of the athletes (*Armstrong, Barker & McManus, 2015*). The degree of biological maturity affects nearly every aspect of a young athlete's physical fitness, including aerobic and anaerobic capacity, strength, and other physiological characteristics (*Abbott et al., 2021*). Incorrect consideration of this factor may lead to inappropriate demands on young athletes and an inaccurate assessment of their potential to achieve results in adulthood (*Cortesi et al., 2020*). In the present study, the authors also evaluated changes in aerobic capacity in both the experimental and control groups. The results suggested that the 3-year training programme contributed to an improvement in aerobic capacity, which may become a crucial factor in achieving high performance in the future (*Chatard et al., 1990*; *Mandigout et al., 2002*; *Oliveira et al., 2021*). Similarly, the eta$^2$ statistic and Cohen's d effect size suggesting large effect of swimming training on aerobic capacity and also indicating moderate to large reductions in body fat at different measurement sites. This conclusion should be approached with some caution because VO$_2$ max was measured using an indirect method. Additionally, the programme did not hinder natural biological development, suggesting that swimming is an appropriate sport for children in the prepubescent stage (*Cortesi et al., 2020*).

Our research demonstrated that swimmers, despite the absence of preliminary selection, had a lower percentage of body fat compared to the control group at the beginning of the study, and this difference persisted throughout the 3-year observation period. However, it should be noted that the control group, who did not engage in swimming, showed a greater increase in the percentage of body fat. This suggests that systematic aerobic exercise in water may inhibit the increase in body fat, consistent with other studies (*Roelofs et al., 2017*; *Nugent et al., 2017*; *Amaro et al., 2017*). On the other hand, some studies indicate that
10-year-old swimmers had a slightly higher percentage of body fat compared to the reference group. By the age of 14, body fat increased in the control group, which the authors attribute to the natural, physiological increase in body fat during adolescence (*Kriemler et al., 2009*). Based on our findings, it can be cautiously concluded that swimming may be an effective means of preventing obesity in young age, considering its low injury rate and numerous health benefits. However, it is worth noting that we did not control the children's diet and other forms of physical activity during the experiment, which may influence the results, although the trend of reduced body fat in the experimental group is evident. Reducing the percentage of body fat may also translate into better sports performance at specific distances and in selected swimming styles. In modern times, in addition to motor and endurance preparation, sports performance depends on many additional factors. It is important to consider other elements that may influence swimming efficiency, such as proper body positioning in water. Even minor changes in body alignment can significantly increase water resistance—the more vertical the body position (greater flexion at the hip, knee, and ankle joints), the greater the resistance. If an athlete does not maintain an appropriate streamlined silhouette, they will have to exert more energy to achieve the desired time (*Hochstein & Blickhan, 2014*).

Another important factor is the level of body immersion. The deeper the body is submerged, the greater the water resistance, which requires greater energy expenditure. Changes in body mass and the percentage of its components directly affect the precision of depth control, maintenance of a streamlined silhouette, and reduction of resistance during swimming (*Segal, 1996*; *Moon, 2013*). It is suggested that a slim physique is desirable in swimming and should be considered during the preliminary selection of children. A slim body build helps to reduce water resistance, which is crucial for achieving good results (*Bartoń, 2024*). Research findings indicate that both the percentage of body fat and its distribution at various anthropometric points in girls aged 10–12 years had a substantial impact on swimming performance. It can therefore be assumed that during the prepubescent period, when athletes begin specialised swimming training, besides technical skills and motor abilities, the percentage of body fat will be a crucial factor influencing the improvement of sports results (*Doherty & Dimitriou, 1997*; *Shepherd et al., 2023*).

The percentage of body fat plays a particularly important role in underwater swimming, which is of great importance, especially over longer distances. It has long been known that to be competitive in local and international competitions, a swimmer must be effective at swimming underwater (*Yustres et al., 2021*). Proper body positioning in the so-called 'torpedo position' after the start and turn is crucial for efficient underwater swimming. Therefore, swimmers should have a low body fat content to reduce drag during underwater swimming. Optimal body positioning, a well-fitted competition suit, and appropriate body composition can significantly improve sports performance, particularly over longer distances (*Lätt et al., 2009*). Body fat content significantly affects aerobic capacity in endurance sports. Cognitive and physical function deterioration, which occurs due to glycogen depletion, can arise even in the presence of excess energy stored as body fat, which the athlete's body cannot efficiently access (*Volek, Noakes & Phinney, 2015*). Other researchers have shown that excessive body fat and increased body weight in prepubescent

children are not necessarily associated with reduced maximal oxygen consumption capacity (*Cornelissen et al., 2011*). However, excess body fat adversely affects submaximal aerobic capacity (*Goran et al., 2000*). In a study conducted by *Marta et al. (2013)*, the authors assessed body fat in 125 healthy children using the Slaughter method, subjecting them to endurance training twice a week for 8 weeks. It was found that higher levels of endomorphy reduced the likelihood of improving vertical jump height, increased mesomorphy promoted better sprint results, and higher levels of ectomorphy increased the likelihood of improving aerobic capacity.

In our research, initial measurements indicated a strongly negative correlation between the percentage of body fat and aerobic capacity in the experimental group. As observations progressed, the impact of body fat on aerobic capacity in swimmers gradually diminished, whereas in the control group this relationship intensified. In the final measurement, the correlation between the percentage of body fat and aerobic capacity was not statistically significant in either group. The authors suggest that endurance training in the swimmer group may have gradually shaped their physique in line with the demands of this discipline, reducing body fat levels and its impact on aerobic capacity. The authors of other studies have proven that endurance training in children contributed to more effective functioning of oxygen enzymes, *i.e.*, succinic acid dehydrogenase (SDH) and citrate synthesis (CS) (*Minasian et al., 2014*). These changes directly affect the increased use of fats as energy substrates and thus the saving of carbohydrates. Therefore, as a result of increasing the use of fat tissue, a greater amount of glycogen remains in the muscles, which affects their more efficient work. Muscle glycogen depletion is one of the most important causes of fatigue during long-term exercise. In relation to the present study, the reduced body fat content in the group training for 3 years may also be related to its more efficient use during exercise, as well as to better $VO_2$ max results. However, this hypothesis requires cautious interpretation, as the final results lacked a statistically significant relationship between body fat percentage and physical fitness. It should also be taken into account that the group size was limited and the diet of the study participants was not taken into account. The sample size was relatively small, which may limit generalizability (pilot study). Further research is necessary in this area. We suggest that swimming instructors and coaches might consider candidates' slim physiques during initial selection, especially regarding the efficiency of underwater swimming.

## CONCLUSIONS

In conclusion, the following cautious statements can be formulated:

Swimming during the prepubescent period may be an effective means of preventing obesity while providing numerous health benefits associated with regular physical activity.

Coaches should consider a slim physique in swimming candidates, as a low body fat percentage favours rapid underwater swimming. However, further research on the influence of body fat in prepubescent children is necessary, also in the context of other sports disciplines.

In long-term studies, a 3-year swimming training programme as an additional form of physical activity resulted in a reduction in body fat growth in all four measurement

locations. At the end of the study, the percentage of body fat was significantly lower in the group of swimmers compared to the group that did not engage in additional physical activity. It can therefore be concluded that the lack of additional sports activity in girls aged 10–12 years contributes to a higher percentage of body fat than in trained girls. Further research in this area in a larger population could provide information on whether the body fat level in untrained girls is within the norm or already indicates obesity.

The initial body fat content had a strongly negative impact on $VO_2$ max, especially in the experimental group; however, in subsequent measurements, this impact diminished and became statistically insignificant in both the experimental and control groups.

### Funding
The authors received no funding for this work.

### Competing Interests
The authors declare that they have no competing interests.

### Author Contributions
- Mariusz Kuberski conceived and designed the experiments, performed the experiments, analyzed the data, authored or reviewed drafts of the article, and approved the final draft.
- Agnieszka Musial conceived and designed the experiments, analyzed the data, prepared figures and/or tables, and approved the final draft.
- Maciej Choroszucho analyzed the data, prepared figures and/or tables, and approved the final draft.
- Jan M. Konarski analyzed the data, authored or reviewed drafts of the article, and approved the final draft.
- Jacek Wąsik conceived and designed the experiments, authored or reviewed drafts of the article, and approved the final draft.

### Human Ethics
The following information was supplied relating to ethical approvals (*i.e.*, approving body and any reference numbers):

Commission relating to Bioethics of Scientific Research Jan Długosz University of Częstochowa.

### Data Availability
The raw data are available in the Supplemental File.

### Supplemental Information
Supplemental information for this article can be found online at http://dx.doi.org/10.7717/peerj.19456#supplemental-information.

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
