# Peer review of "Longitudinal effects of aerobic training programme on body composition in non-elite adolescent female swimmers"

_PeerJ, doi:10.7717/peerj.19456_

## Round 0.1 · original submission · Major Revisions

Dear Authors,

Please revise the manuscript considering the reviewers´ suggestions.

Thank you.

Best regards.

·

Basic reporting

The English must be carefully checked again.
The literature is keen on the topic.
The tables and figures need under them captions, describing abbreviations and indexes.
The Hypothesis - Results and Discussion have been connected. However, it is important to include the limitations of the study.

Experimental design

My only concern is about the methodological limitations that I have clarify in the general comments.

Validity of the findings

The authors must eliminate the probability that the limitations will affect the results.

Additional comments

28. It is better to refer on the title as “Body composition”
99. I suggest to include subtitles for each part of the introduction.
113. fetus
162. Compromising
164-165. Did you check the attendance, performance, or health issues before their participation? I believe that you should include inclusion and exclusion criteria for both groups.
179. Refer to the number of measurements.
182. Can you use an additional reference to regard this formula?
194. the body fat percentage
203. This must be one of the limitations of your study. I understand that the control group could not swim, however, this test probably ameliorates the swimmers’ aerobic fitness. It could be more appropriate to use a test in water, providing probable correlations with the Beep test.
210. meters
216. It is also a limitation of your study that you must refer to. The indirect calculations of VO2max do not provide precisely the variable.
218. delete “in”
224. Correct “meters” where it is necessary.
Include the type of your study’s design. Also, refer to the number of measurements throughout the three years. Did you check the variables in the off-season periods? Moreover, did you track the World Aquatics points to clarify the swimmers' level?
227. Refer to the probable analysis for normality and the probability that you calculated the sample size with a priori sample size calculation.
263. Did you observe the menarche throughout the measurements?
300. Delete “interestingly”. Please, do not discuss your results in that section.
318-319. Please, extend your comment. I do not understand what you mean.
328. This sentence is not clear. The technical proficiency depends on the anthropometrics. Check this reference: 38. Przewęda, R. Ocena wieku rozwojowego. In Teoria i metodyka sportu; Ulatowski, T., Ed.; Sport i Turystyka: Warszawa, 1971.
342. On the other hand, this means that school physical education lessons are not able to provide any physiological or body composition improvement.
You must include a limitations section.

Reviewer 2 ·

Basic reporting

see file

Experimental design

see file

Validity of the findings

see file

Additional comments

see file

Annotated reviews are not available for download in order to protect the identity of reviewers who chose to remain anonymous.

Reviewer 3 ·

Basic reporting

The background and the supporting themes provide a broad context but do not fully convince about the impact of this study (i.e. the use of a swimming intervention as a model to study body fat changes of young girls). However, impact is not a concern for PeerJ. I evaluated the work based on the scientific rigour of the study and there are several methodological flaws.

The writing overall shows some parts that are ambiguous and/or lack clarity. The figures and tables may be informative but for an applied field (coaches). The 6 time points displayed would benefit from a graphical experimental timeline showing when 6 time points occurred over time.

The authors explained that they wanted to show how the experimental and control groups change in terms of body fatness over a period of 3 years in a situation when respondents were not selected for the study. The current title does not seem to clearly highlight this main fact (as stated in the cover letter) that respondents were not selected for the study (i.e. without preliminary selection).

Experimental design

The research was conducted in conformity with the prevailing ethical standards in the field that I could verify. But the experimental design and methods are not scientifically rigorous and show poor technical standard. Especially for the monitoring of the intervention programme of the experimental group. It was assumed that all participants adhered to the British Swimming guidelines and a general description only was provided (not enough for replication).

Another example of poor scientific rigour is the description of skinfold measures that do not match the parameters required for the calculation of body fat.

Inclusion and exclusion criteria should be more explicitly and clearly described. No sample size justification, n = 10 appears arbitrary and likely small, reliability of the dependent variables is not presented. Who and how many researchers tested participants over 3 years is not known. Another replicability issue.

Validity of the findings

I am unable to trust data where the independent variable was manipulated under the assumption that the participants performed a comparable training intervention over time, with no even minimal monitoring to verify adherence, and key training variables such as intensity and volume.

Normal distribution of dependent variables is based on visual inspection rather than the typical normality test (Shapiro-Wilk). Also, even when visually looking at the histograms not all data appear normally distributed. Following Shapiro-Wilk (Jamovi 2.3.28.0) spot checks, some datasets of the experimental group appear not normally distributed. As an example: stomach (mm) 1 ( p = 0.021); wiek biol.(lata) 4 ( p = 0.004); biceps (mm) 1 ( p = 0.029). These are all non normally distributed variables requiring non-parametric statistics.

For a robust analysis and interpretation of findings (beyond the assumption of a rigorous methodology and an appropriate sample size) it is expected the inclusion of the effects size, and/or confidence intervals. Without these parameters, the robustness of the analysis and so conclusions drawn from the data can be misleading.

Additional comments

Concise sample of minor and major concerns.

Line 69: If we refer to “well-established by research”, I think a meta-analysis study is more appropriate and could be added to support any statement.
Line 73: the flow of writing is poor here. I think a transition should be added. For example: “…biological development [4,7]. Swimming, a low-impact sport with numerous benefits, is an ideal choice to meet these guidelines. Numerous..” or a similar transition.
Line 87: why?
Line 83 to 87: It reads like independent notes that are not fully contributing to the argument for the rationale.
Line 94: does it refer to swimmers, active individuals or sedentary? It is not clear.
Line 102-104: I’m not clear overall with this paragraph, especially when stating “even among girls of normal body weight”. Excessive fat accumulation means overweight and/or obesity instead?
Line 114: this statement requires citations. Authors who?
Line 112-122: This part should be reorganised because it starts with an obesity argument, move to swimming, move back to obesity, and add a theme about diet. In the context of the rationale of the study it reads confusing.
Line 141: statement requiring citations.

Line 161: typically, the age is a characteristic reported into brackets like hight and body mass, were all participant 10 years old?
Line 167: what is the age of the control group?
Line 162: why 14 participants? No sample size justification, no demonstration of appropriate statistical power for data analysis.
Line 180: report the stadiometer details as it is expected for any instrument used for a measure. Another example of lack of details showing poor scientific writing rigour.
Line 189: why 4 anatomical sites? Is it valid performing the assessment of body fat with this approach? No evidence provided. Why measure over the biceps and at the iliac crest but the formula did not use them? R represents the skinfold measurement over the triceps, L represents the skinfold measurement beneath the scapula.
Line 204: what is the evidence showing that the test is an accepted way to measure the aerobic power performance for the selected population? No reference to previous works.
Line 221-225: it is assumed that the programme adhered to the theoretical plan, but was any form of training (intensity/volume/perception of effort/etc) quantified over time to prove it? This is a major flaw because there is an assumption of comparable training among participants of the experimental group.

---

## Round 0.2 · Major Revisions

The reviewers again pointed out important aspects to improve in the article. Please consider what the reviewers suggested in the initial review and in round 1.

Thank you.

Best regards.

·

Basic reporting

The authors tried to improve their manuscript. However, there are many limitations, considering the way that the samples measured on an important factor such as aerobic capacity.

Experimental design

228. My previous comment (203 line) was about the validity of a test outside the water in aerobic capacity parameters. How can you assess the aerobic fitness of a swimming program with a dryland test? Why did not use a safe in water test that will ensure the adaptability of your sample and it will give a clear result about the specific adaptation of aerobic fitness in the water?
193. How did you examine their premenarchal condition. Please, extend the paragraph, describing the methodology that you used.

Validity of the findings

268-269. Extend your limitation section. Refer to details about the reasons that you selected this test and you had a small sample size.

Additional comments

70. The effects of swimming in what? Please, define.
230 meters

Reviewer 2 ·

Basic reporting

no comment

Experimental design

Regarding my previous comment:
"What about effect sizes? For the ANOVA, and pairwise comparison?"

You answered:
"Effect sizes are presented in Table 2 as F"

This is wrong! The effect size for the ANOVA is the eta square, and for the pairwise comparison the Cohens's d. This needs to be revised, and also the results interpretation based on such values.

Validity of the findings

no comment

Additional comments

no comment

---

## Round 0.3 · Major Revisions

Dear Authors,

Please revise the manuscript considering the reviewers´ notes.

Thank you.

Best regards.

·

Basic reporting

The authors

Experimental design

In my point of view, I believe that the methodology has significant flaws; therefore, I suggest to debate their results considering this limitation (VO2 examination).

Validity of the findings

A VO2max test using kicks, for both groups, would be a better solution, avoiding the limitation of a dryland test.

Additional comments

I appreciate the authors' efforts to explain their point of view. However, I still believe that the methodology has many limitations. Also, I suggest to modify the construct of the manuscript placing limitations in a summarizing paragraph at the end of discussion.

Reviewer 4 ·

Basic reporting

The manuscript is well-structured and written in clear and professional English. However, the title could be more precise—the phrase "not selected for the sport of swimming" is uncommon in the literature. A clearer alternative could be "non-elite adolescent swimmers" or "adolescent swimmers not undergoing competitive selection."

Additionally, the abstract should emphasize the main findings more clearly. The limitation of using a dryland aerobic test should also be briefly mentioned to provide context for the reader.

Minor formatting issue: Ensure that p-values are formatted consistently throughout the manuscript (e.g., p = 0.005 instead of p<0.05).

Experimental design

Aerobic Capacity Assessment on Land Instead of Water:
A previous concern was raised regarding the validity of using a dryland test to assess aerobic capacity improvements in swimming training. The 20m shuttle run test does not fully capture swimming-specific adaptations. While the authors justify this decision based on the control group’s inability to perform an in-water test, this does not fully resolve the issue of ecological validity.
→ Suggestion: The authors should explicitly acknowledge this as a limitation and strengthen their justification by referencing studies that successfully conducted in-water aerobic tests in similar populations.

Premenarchal Status Determination:
The method used to determine maturation status remains questionable. It was assessed through parental interviews, which is subject to recall bias and lacks biological confirmation.
→ Suggestion: The authors should include references to studies that use more objective anthropometric-based methods (e.g., PHV calculation) for assessing maturation, or address this as a major limitation.

Sample Size Justification:
The limitations section states that the sample size estimation was performed using G*Power, but it is unclear whether this was done a priori or post hoc.
→ Suggestion: The authors should clarify whether the power analysis was conducted before or after data collection, as this impacts the robustness of the sample size justification.

Potential Confounding Variable Not Acknowledged:
The limitations section should explicitly acknowledge that the study did not control for unsupervised physical activity in the control group. This is a key confounder that may have influenced body composition and aerobic fitness outcomes.

Validity of the findings

ffect Sizes:
The manuscript reports eta squared (η²) for ANOVA and Cohen’s d for pairwise comparisons, which is appropriate. However, the interpretation of these effect sizes should be checked against established benchmarks (small, medium, large effects) to ensure meaningful contextualization.

Linear Mixed-Effects Models (LMM):
The assumptions of the LMM model are not discussed. Were normality of residuals and random effects structure verified? Were random slopes tested in addition to random intercepts? A statement on these aspects should be added for transparency and reproducibility.

Interpretation of Results (VO2max Changes Over Time):
The discussion regarding the relationship between body fat and VO2max over time lacks depth. The manuscript states that body fat percentage initially impacted VO2max, but this effect diminished over time, but it does not explain why this happens.
→ Suggestion: The discussion should include additional references to physiological mechanisms explaining this trend.

Confounders Not Fully Addressed:
The following potential confounding factors remain unclear:

Dietary intake – Was diet controlled or monitored?
Training outside the study – Did the control group engage in unsupervised physical activity?
Maturational differences – Were growth spurts considered as a possible factor in body composition changes?
Discussion Structure:
The lack of differences in body fat percentage at baseline and the significant divergence over time should be analyzed in greater depth.
→ Suggestion: The authors should consider whether natural growth and development played a role in these changes. Could weight gain in the control group be attributed to lower overall physical activity levels?

Additional comments

This study presents interesting findings regarding the effects of swimming training on body composition in adolescent females. However, key methodological and analytical issues need to be addressed before publication.

Major areas requiring revision:

The ecological validity of the dryland aerobic test should be explicitly acknowledged as a limitation.
Sample size justification must clarify whether G*Power analysis was conducted a priori or post hoc.
The lack of an objective anthropometric-based maturation assessment (e.g., PHV calculation) should be acknowledged in the limitations section.
The study does not control for unsupervised physical activity in the control group, which could bias results—this needs to be mentioned as a limitation.
The discussion should provide a more detailed explanation of why body fat had a decreasing impact on VO2max over time, supported by references to physiological mechanisms.
LMM assumptions were not discussed, and their validation should be explicitly stated.

While the study offers valuable insights, addressing these methodological, analytical, and confounder-related concerns is necessary to strengthen the validity of its conclusions.

---

## Round 0.4 · Minor Revisions

Dear Authors,

Thank you for your work during the review process.

Please consider the feedback by reviewer 2.

Thank you.

Best regards.

**Language Note:** The review process has identified that the English language must be improved. PeerJ can provide language editing services - please contact us at [email protected] for pricing (be sure to provide your manuscript number and title). Alternatively, you should make your own arrangements to improve the language quality and provide details in your response letter. – PeerJ Staff

Reviewer 2 ·

Basic reporting

The authors have already addressed my comments in the last version of the manuscript.

Experimental design

The authors have already addressed my comments in the last version of the manuscript.

Validity of the findings

The authors have already addressed my comments in the last version of the manuscript.

Additional comments

The authors have already addressed my comments in the last version of the manuscript.

·

Basic reporting

The manuscript is clear and written in a professional English. Appropriately terminology is used, as well the structure of the article, with logical flow and good organization of figures ans tables. Table 2 gives comprehensive descriptive statistics and results from ANOVA tests, alongside with the effect size indicators, supporting the statistical analysis.

However, I suggest some improvements:
Attachments:
Table 1. Descriptive statistics of the first and sixth measurement timepoint, along with results of the ANOVA, should be mentioned Table 2.

There appears to be a spelling mistake in "skrew" (should be "skew").

Suggestions for improvement: standardization of formatting, clarifying the terms used in the table to facilitate interpretation.

Experimental design

The experimental design meets the standards of original primary research within the aims and scope of the journal. The research question is well-defined, meaningful and clearly fills a knowledge gap by focusing on non-elite adolescent swimmers who have not been subjected to preliminary selection.

Topics for improvement:
. The fairly small sample size (n=14 per group) limits the generalization of results. Tought a powerfull calculation is provided.

. The study didn't monitorized control dietaty intake or other physical activities outside of swimming or physical education classes. However is mentioned as a limitation, these factores could dratically influence body composition and aerobic capacity.

. The support on the shuttle run test may not fully reflect swimming specific aerobic capacity.

In conclusion this manuscript is a well executed longitudinal study that offeres valuable insights.

Validity of the findings

Overall, the findings are valid and well-supported by the data and analyses. The study demonstrates careful methodological planning and transparent reporting.

It would be advantageous to make sure that the raw data are readily available in a manner that can be used for replication or reanalysis, even though the publication claims that all data were supplied (e.g., CSV, annotated Excel). This enhances the study's reproducibility and transparency.

Although the study's justification is obvious, it would be beneficial to replicate the results in other populations (such as males, competitive swimmers, or people from diverse cultural backgrounds) in order to increase the findings' applicability.

The study's conclusions are logically and statistically sound, supported by good methodology, and directly related to the objectives that were stated. The findings have a high degree of validity, provided that data sharing is slightly improved and the implications are more widely contextualized.

Additional comments

A significant and little-studied topic is covered in this well-organized and lucid manuscript: the long-term impacts of swimming on body composition in non-elite, prepubescent adolescent females. The authors deserve praise for employing suitable statistical methods and for carrying out a three-year longitudinal study with regular measurement intervals.
The limitations are openly addressed, and the discussion section effectively links the findings to previous research. Value is added to applied settings by focusing on the real-world consequences for educators and coaches.

Nevertheless, the following editorial and stylistic changes would improve readability:
There are a few small typographical and grammatical problems throughout, including inconsistent usage of symbols like "±" and "%," incorrect punctuation, and spacing faults. A thorough copyedit or proofread is advised.
There is considerable ambiguity in the term "selection" when it comes to athlete recruitment. For an international readership, it would be more clear if it were made clear if this was referring to the recognition of talent or just voluntary involvement.
To increase reader involvement and flow, think about condensing or rearranging parts of the discussion's larger paragraphs.
All things considered, this is a solid book that could make a significant addition to the body of knowledge about teenage health and athletic engagement.

---

## Round 0.5 · accepted · Accept

Dear Authors,

Thank you and congratulations on your work during the review process, addressing all of the reviewers' comments.

Best regards.

·

Basic reporting

In terms of the changes requested for the basic reporting field, table 1 has been adjusted and spelling errors corrected. Formatting issues and clarification of the terms used in the tables have also been corrected. Issues such as the lack of monitoring of diet control, extra-curricular physical activities and sample size were well justified and accepted.

Experimental design

In terms of experimental design, all the issues raised for improvement were well justified and framed within the scope of the study.

Validity of the findings

In terms of the Validity of the findings, the discussion has been improved for smaller chapters in order to make it easier for the reader to follow and understand. Some of the terms used have also been clarified and changed.